# Unveiling Insights: A Bibliometric Analysis of Artificial Intelligence in Teaching

**Malinka Ivanova** [1,*] , **Gabriela Grosseck** [2] **and Carmen Holotescu** [3]

1 Department of Informatics, Faculty of Applied Mathematics and Informatics, Technical University of Sofia, Blvd. Kl. Ohridski 8, 1797 Sofia, Bulgaria
2 Department of Psychology, Faculty of Sociology and Psychology, West University of Timisoara, 4 Bd. Vasile Pârvan, 300223 Timisoara, Romania; gabriela.grosseck@e-uvt.ro
3 Department of Information Technology, Faculty of Engineering, "Ioan Slavici" University of Timisoara, 144 Str. Păunescu Podeanu, 300569 Timisoara, Romania; carmen.holotescu@islavici.ro
* Correspondence: m_ivanova@tu-sofia.bg

**Abstract:** The penetration of intelligent applications in education is rapidly increasing, posing a number of questions of a different nature to the educational community. This paper is coming to analyze and outline the influence of artificial intelligence (AI) on teaching practice which is an essential problem considering its growing utilization and pervasion on a global scale. A bibliometric approach is applied to outdraw the "big picture" considering gathered bibliographic data from scientific databases Scopus and Web of Science. Data on relevant publications matching the query "artificial intelligence and teaching" over the past 5 years have been researched and processed through Biblioshiny in R environment in order to establish a descriptive structure of the scientific production, to determine the impact of scientific publications, to trace collaboration patterns and to identify key research areas and emerging trends. The results point out the growth in scientific production lately that is an indicator of increased interest in the investigated topic by researchers who mainly work in collaborative teams as some of them are from different countries and institutions. The identified key research areas include techniques used in educational applications, such as artificial intelligence, machine learning, and deep learning. Additionally, there is a focus on applicable technologies like ChatGPT, learning analytics, and virtual reality. The research also explores the context of application for these techniques and technologies in various educational settings, including teaching, higher education, active learning, e-learning, and online learning. Based on our findings, the trending research topics can be encapsulated by terms such as ChatGPT, chatbots, AI, generative AI, machine learning, emotion recognition, large language models, convolutional neural networks, and decision theory. These findings offer valuable insights into the current landscape of research interests in the field.

**Keywords:** artificial intelligence; teaching; intelligent environment; learning analytics; large language models; ChatGPT

## 1. Introduction

Integrating AI into teaching represents a rapidly evolving field with a wide array of research and developments [1,2]. As the integration of AI in educational contexts continues to grow [3], conducting a bibliometric study on the topic becomes essential for several reasons [4], including:

*Rapid Growth in AI in Education.* The field of AI in education has witnessed significant advancements and innovations in recent years [5]. With the growing emphasis on personalized learning, adaptive teaching methodologies, and the integration of AI technologies in classrooms [6], there is a pressing need to assess the current state of research and track emerging trends during the study period 2018–2023.

*Evolving Research Landscape.* Changing Environment for Research. Researchers from a variety of fields, including education, computer science, philosophy, communication, sociology, neuroscience, management and psychology, are involved in the interdisciplinary field of artificial intelligence in education and teaching [7–9]. This is particularly true with regard to research on ChatGPT, since it has been growing quickly and the course and evolution of its future are still very much up in the air. As a result, it becomes crucial to review and update the bibliometric study's findings. According to Farhat et al., these updated studies can also provide insight into the dynamics and development of the scientific community and field that surround ChatGPT research [10]. More bibliometric analysis can assist in delineating the dynamic research terrain, pinpointing significant contributions, and comprehending the cooperative networks within this interdisciplinary domain.

*Evaluation of Research Impact.* Such a study can provide an assessment of the impact and influence of the topic "AI in teaching" publications. Despite AI's increasing integration into various academic domains, from teaching and learning to research methodologies, systematic investigations into its broader implications remain scarce. For example, in their systematic review, Chiu et al. examine the integration of AI into four key educational domains (learning, teaching, assessment, and administration) over the past decade, and suggests future directions for research on the connection between AI technologies and their use in teaching [11]. Thus, by including factors such as citation count, journal impact factors, and author h-index, our study can provide valuable insights into the most impactful studies and influential researchers during the given time frame (2018–2023).

*Identifying Knowledge Gaps.* Despite recent interest in AI in teaching and learning [12], there may still be areas that have received limited attention in the academic literature. Some educational domains where AI hasn't gotten much attention in the scholarly literature include: emotional intelligence [13], the ability to promote kindness and empathy [14,15] and the wellbeing of students, teachers, and other educational stakeholders [16], student creative thinking [17], cultural sensitivity and diversity in education [18], gender issues arising when using ChatGPT [19] or gender-based violence [20], physical education and sports training through personalized coaching and performance analysis [21], special educational needs in personalized learning, communication, and skill development [22], etc. A bibliometric analysis can help identify knowledge gaps and underrepresented topics, guiding future research directions and potential areas of exploration.

*Policy and Educational Decision Making.* Policymakers, educational institutions, and stakeholders increasingly rely on evidence-based research to make informed decisions. Popenici et al. identifies several factors that influence the connection between the higher education teaching process and artificial intelligence [23]. The impact of AI on academic integrity and student learning, the opportunities and challenges of integrating AI into the current curriculum and assessment frameworks, the ethical, social, and legal ramifications of using AI in education, and the pedagogical and epistemological presumptions underlying AI systems are some of these factors. In order to guarantee that AI is applied responsibly and profitably for higher education, the authors make the case that researchers, educators, and legislators must carefully analyze and address these problems. On the other hand, a more inclusive approach to address the originality of students' work is required, as Luo argues [24]. Thus, a comprehensive bibliometric study can serve as a reliable source of information for shaping policies related to AI integration in teaching, promoting best practices, and optimizing resource allocation.

*Benchmarking Progress.* The period from 2018 to 2023 likely saw notable advancements in AI technologies and their utilization in teaching. Authors such as Talan [25], Li and Wong [26] and Maphosa & Maphosa [27] delved into the transformative potential of AI in teaching, contributing to the bibliometric analysis of literature on AI usage in education. Their work utilized bibliometric analysis and topic modeling techniques to provide insights into the evolving landscape of AI's impact on teaching methodologies and learning outcomes within educational settings. As such, a bibliometric study can serve as a benchmark

to gauge the progress made during this time span and provide valuable insights into the trajectory of AI in teaching research.

*Global Perspective.* Both Scopus and Web of Science (WoS) are reputable databases that index articles from a big number of international journals and conferences in different domains [28,29]. By using these databases as the data source, the bibliometric study can provide a global perspective, encompassing research contributions from various countries and regions.

Moreover, since November 2022, the rise of ChatGPT and other generative AI models has further amplified the significance of conducting a bibliometric study focusing on AI in teaching. As ChatGPT and similar models become increasingly prevalent in educational contexts [30,31] understanding the research landscape and the impact of AI in teaching becomes even more imperative. These powerful language models are characterized by the possibility to revolutionize educational practices by offering personalized tutoring, generating interactive learning content, and assisting teachers in designing more effective instructional materials [32–34]. Despite recent contributions such as those by Dempere et. al., which offer a comprehensive analysis of ChatGPT's scholarly footprint encompassing publication trends, citation patterns, collaborative networks, and application domains [35], the existing literature remains relatively sparse. Subjects pertaining to ChatGPT are dispersed across various fields of research. For instance, Barrington et al. conducted a bibliometric analysis focusing on ChatGPT literature within the realms of medicine and science [36]. A comprehensive bibliometric analysis would not only shed light on the trajectory of research in this field but also capture the influence of generative AI models on teaching methodologies and educational outcomes. By including these innovative AI technologies in the study, researchers and educators can get valuable knowledge into the evolving landscape of AI in teaching and its potential to reshape the future of education.

Nevertheless, upon reviewing the available literature, it came to our attention that there has been limited coverage of bibliometric analysis regarding AI in teaching within scientific publications. However, a number of recent studies have commenced addressing this particular gap. One such study by Crompton et al. [37] examined the research landscape of AI integration in K-12 classrooms. Their analysis revealed an increasing trend in publications over the past decade, highlighting the growing interest in AI's impact on teaching practices. Additionally, a study by Liang et al. [38] explored the implementation of AI-driven virtual reality in language teaching and learning. Their analysis uncovered emerging trends and highlighted the potential of AI-powered VR applications in providing immersive language learning experiences. The recent bibliometric analysis of Polat et al. contributes to a comprehensive understanding of the current state of ChatGPT research in education, offering researchers and practitioners valuable insights into evolving trends and potential future directions for this innovative aspect of AI and learning [39]. These bibliometric studies collectively contribute to a deeper understanding of AI's role in teaching, identifying research trends, knowledge gaps, and future research directions in this rapidly evolving field. As more researchers recognize the significance of artificial intelligence in education, we can expect an increase in bibliometric studies to further enrich our insights into this transformative area of study.

It can be said that while focusing on AI in teaching provides valuable insights into the influence of AI on teachers and instructional practices, AI's position in the learning process and research is also critical. Researchers interested in a more holistic understanding of AI's influence on education may consider conducting separate studies or expanding their investigation to encompass AI in teaching and learning as separate components.

Therefore, the purpose of this work is to undertake a bibliometric study based on Scopus and Web of Science research for the year 2018–2023 in order to outdraw and analyze the worldwide picture regarding the utilization of AI in teaching practice. Given the dynamic and revolutionary nature of AI in the educational domain, it is highly merited and requested. Thus, we address several *research questions*:

- *Publication trends.* What is the annual scientific publication growth? Which are the most productive countries? Which journal do scholars mostly publish in? What are the most relevant affiliations? Which authors are the most productive?
- *Citation analysis.* Who are the most cited scientists and scholars? What is the academic performance of the AI in teaching theme in the Scopus and WoS database? Is there a certain level of authors' contribution that follows a particular pattern?
- *Collaborative networks.* Which countries collaborate in AI in teaching research? What is the specific contribution pattern of authors who researched this topic?
- *Application domain and future directions.* What is the conceptual structure of the research field? What are the most relevant topics in the research developed on AI in teaching? How has the research progressed over the past 5 years?

As the primary goal of this paper is to conduct a bibliometric analysis of AI in teaching research, we have established the following specific objectives, to accomplish this:

*Establishing a descriptive structure* of the scientific production through obtaining annual growth, the number of indexed documents in Scopus and Web of Science during the investigated period, number of authors, countries, institutions and publication sources.

*Determining the impact of scientific publications* through information mining regarding average citation per document, most cited countries considering two different parameters: total citations and average article citations.

*Tracing collaboration patterns* considering the number of co-authors per document, as well as the formed authors, institutions and country collaboration networks.

*Identifying key research areas and emerging trends* considering the frequently used author's keywords to describe in the best way the paper content.

By addressing these research objectives, we are convinced that such an analysis will contribute significantly to the development and implementation of AI technologies in education and, ultimately, improve learning experiences for students worldwide.

The structure of the paper is as follows: After the Introduction, we continue with Section 2, which describes the methodology of the study. Section 3 is dedicated to analyzing data and results, followed by a discussion in Section 4. In Section 5, we draw several conclusions.

## 2. Methodology

### 2.1. Methods and Tools

The bibliometric analysis is conducted according to bibliographic data taken from Scopus and the Web of Science scientific database on 1 December 2023. The passed query is related to *"artificial intelligence" and teaching* as the search is conducted in article title, abstract and keywords in Scopus and in All Fields in WoS.

In order to quantify scholarly communication, we worked with Biblioshiny (a bibliometric software package web-based on R language) to analyze and visualize the research status and trends in the field of AI in teaching [40].

### 2.2. Sources and Data Collection

The documents are extracted from the Scopus and WoS databases which are considered some of the most important and comprehensive collections of scientific resources worldwide for detailed bibliometric analysis [41].

We based our search on PRISMA (Preferred Reporting Items for Systematic Reviews and Meta-Analyses) guidelines [42,43]. As a result, on the query, we extracted a total of 18,741 (10,254 from Scopus and 8487 from WoS) documents which were downloaded in a tab separator format. Figure 1 shows the refining process until the final set was obtained.

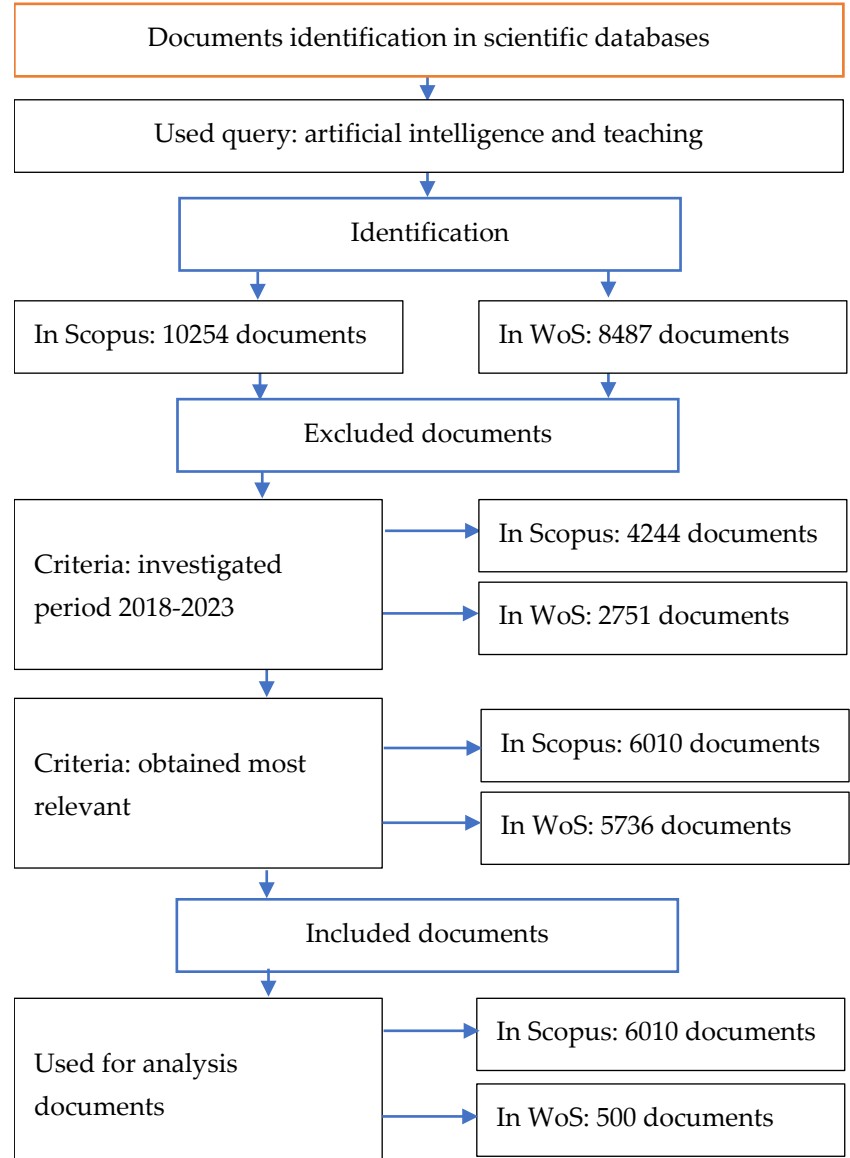

**Figure 1.** The PRISMA process for document collection.

### 3. Data Analysis and Results

Bibliometric analysis is performed considering the formulated main objectives related to the establishment of a descriptive structure of obtained documents, determination of the impact of scientific publications, tracing collaboration patterns, and identification of the key research areas and emerging trends. The findings are summarized to outline some facts, achievements and trending topics concerning AI in teaching.

### 3.1. Establishing a Descriptive Structure

The interest in the researched topic can be judged by the annual scientific production, as it is presented in Figure 2. The curve extracted from Scopus is increasing and the annual scientific production is characterized with an annual growth rate of 25.42 % as for the 2018 year 485 articles are indexed in Scopus, in the 2019 year the documents are 510, 2020 year-839, 2021 year-1261, 2022-1410. For the year 2023, 1505 documents have been indexed up to the time of the research. The curve characterizing the annual scientific production according to WoS is also increasing as the annual growth rate is 39.33%. for the investigated period.

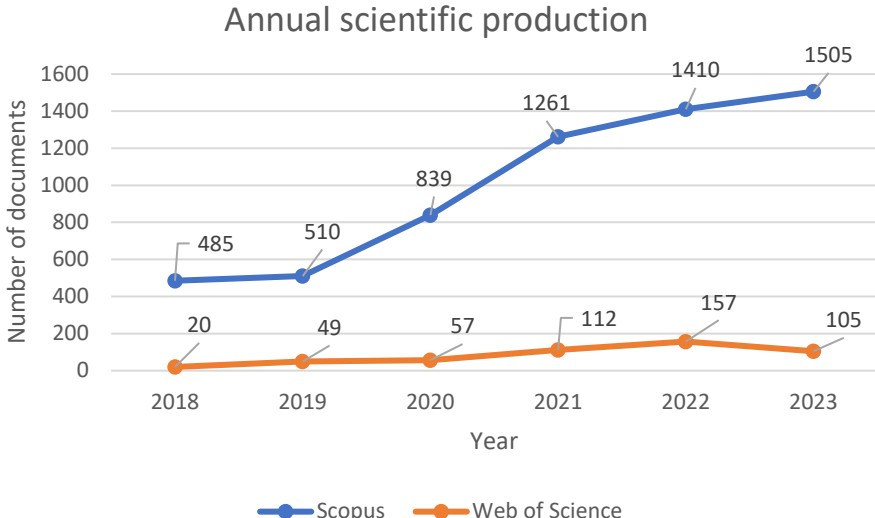

**Figure 2.** Annual scientific production for the period 2018–2023 according to Scopus and Web of Science.

Figure 3 shows the most contributed countries over time, involved in research of this scientific topic. The most active authors according to Scopus are from: China (2023-5446 documents), USA (2023-2074 documents), India (2023-1018), Germany (2023-618), UK (2023-605), Spain (2023-550), Australia (2023-383), Brazil (2023-355), Italy (2023-299), and Malaysia (2023-268). All columns are increasing as the production of China is impressive, with the difference between the first country and the tenth being about 20 times for the 2023 year.

According to WoS, the authors from the following 10 countries most often publish on this topic: China (2023-521 documents), USA (2023-91), Spain (2023-35), India (2023-34), UK (2023-32), Canada (2023-32), Germany (2023-30), Chile (2023-30), Ecuador (2023-24) and Korea (2023-21).

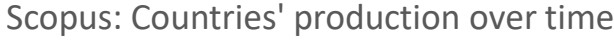

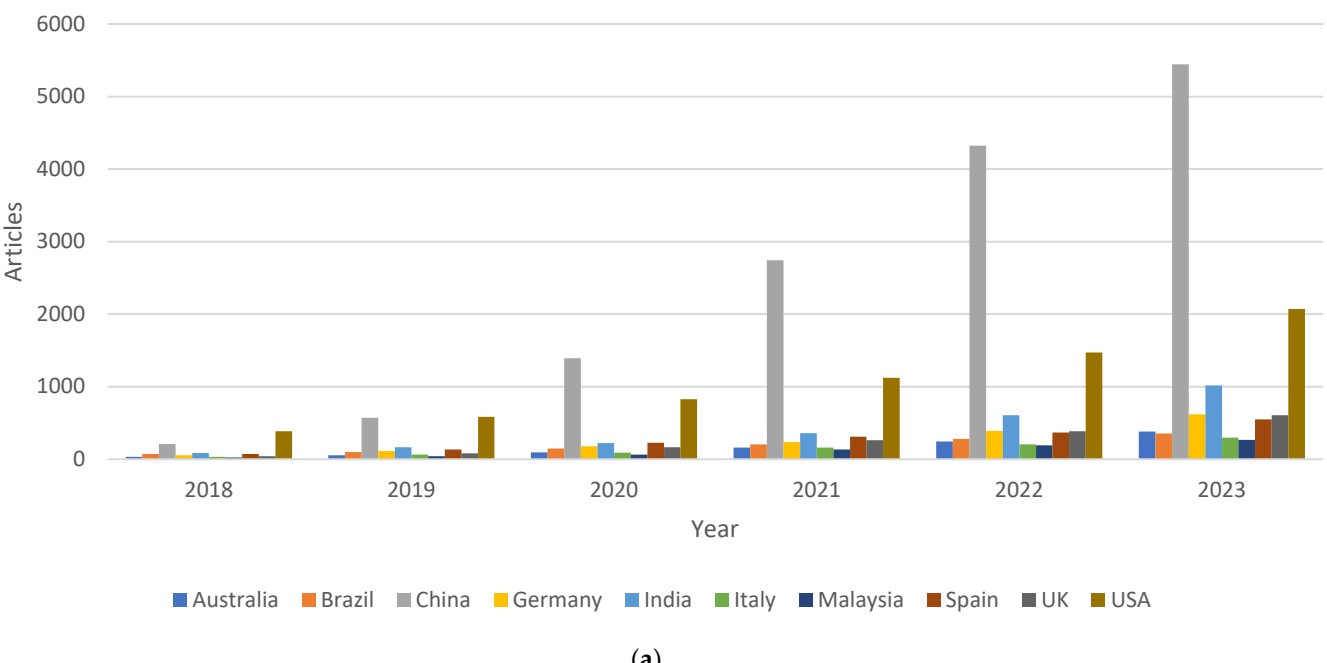

**(a)**

**Figure 3.** *Cont.*

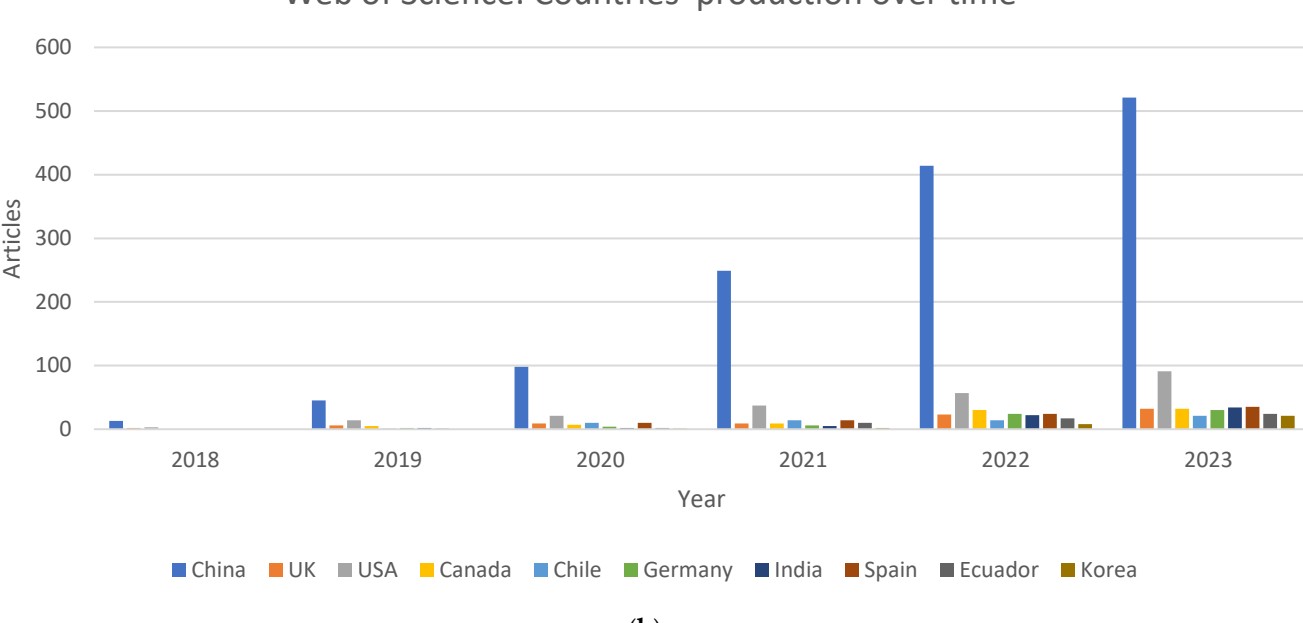

**Figure 3.** Countries' production over time according to (**a**) Scopus, (**b**) Web of Science.

The most relevant sources that publish articles devoted to the investigated topic are presented in Table 1. According to Scopus, it seems that *Journal of Physics: Conference Series* includes and disseminates the bigger part of articles as it characterizes with SJR 2022: 0.18. Other preferred sources are: *ACM International Conference Proceeding Series* (SJR 2022: 0.21), *Lecture Notes in Computer Science* (Q3, SJR 2022: 0.32), *Advances in Intelligent Systems and Computing*, *Lecture Notes in Networks and Systems* (Q4, SJR 2022: 0.15), *Communications in Computer and Information Science* (Q4, SJR 2022: 0.19). In WoS, the most relevant sources are completely different from those in Scopus and the top five are: *Journal of Intelligent & Fuzzy Systems* (Q2, SJR 2022: 0.37), *Mobile Information Systems* (Q3, SJR 2022: 0.36), *Wireless Communications & Mobile Computing* (Q2, SJR 2022: 0.45), *Frontiers in Psychology* (Q2, SJR 2022: 0.89), *Computational Intelligence and Neuroscience*.

**Table 1.** Most relevant sources according to Scopus and Web of Science.

| Source | Number of Published Papers |
|---|---|
| **In Scopus** | |
| *Journal of Physics: Conference Series* | 316 |
| *ACM International Conference Proceeding Series* | 272 |
| *Lecture Notes in Computer Science* (including subseries *Lecture Notes in Artificial Intelligence* and *Lecture Notes in Bioinformatics*) | 267 |
| *Advances in Intelligent Systems and Computing* | 140 |
| *Lecture Notes in Networks and Systems* | 102 |
| *Communications in Computer and Information Science* | 76 |
| *Wireless Communications and Mobile Computing* | 76 |
| *CEUR Workshop Proceedings* | 66 |
| *Mobile Information Systems* | 66 |
| *Journal of Intelligent and Fuzzy Systems* | 62 |
| **In Web of Science** | |
| *Journal of Intelligent and Fuzzy Systems* | 41 |
| *Frontiers in Psychology* | 16 |
| *Computational Intelligence and Neuroscience* | 12 |
| *International Journal of Emerging Technologies in Learning* | 12 |

**Table 1.** *Cont.*

| Source | Number of Published Papers |
|---|---|
| *Mobile Information Systems* | 9 |
| *Sustainability* | 9 |
| *Education and Information Technologies* | 8 |
| *Wireless Communications and Mobile Computing* | 8 |
| *Scientific Programming* | 7 |
| *Lecture Notes in Real-Time Intelligent Systems* (RTIS 2016) | 6 |

Among the most relevant affiliations of contributed authors according to Scopus (Figure 4a) are: Beijing Normal University (China), Central China Normal University (China), South China Normal University (China), Carnegie Mellon University (USA), The University of Hong Kong (Hong Kong), Wuhan University of Science and Technology (China), Monash University (Australia), University of Belgrade (Serbia), McGill University (Canada). In Web of Science (Figure 4b), the similar authors' universities are Beijing Normal University (China), Central China Normal University (China), South China Normal University (China), and McGill University (Canada).

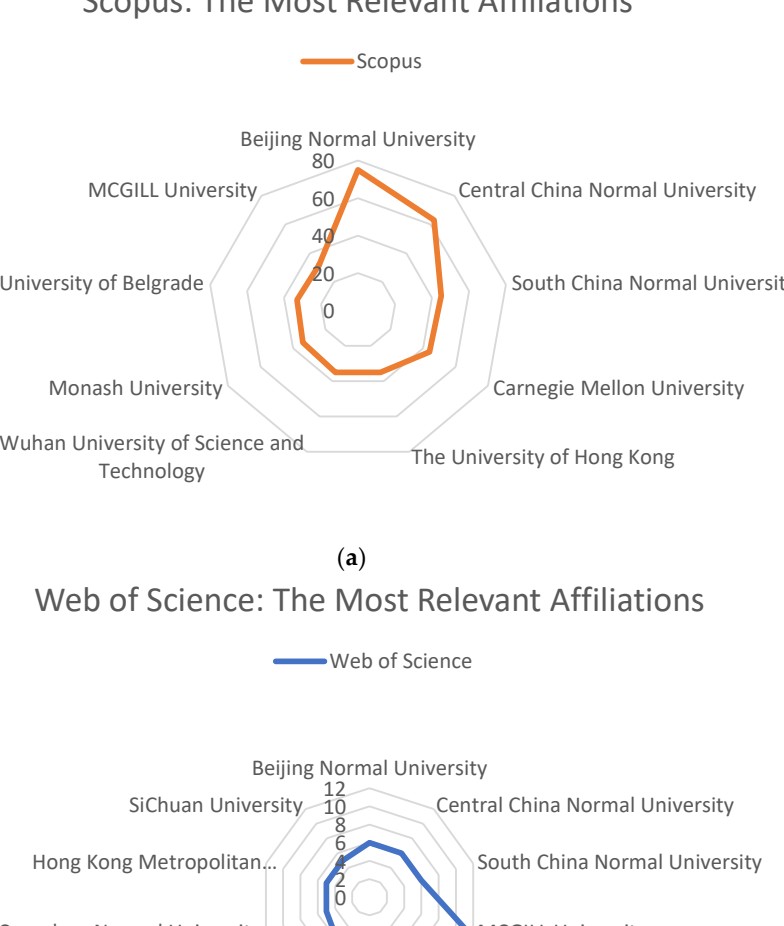

(**a**)

(**b**)

**Figure 4.** The most relevant affiliations according to (**a**) Scopus, (**b**) Web of Science.

The author's impact (h-index) according to Scopus and Web of Science is presented in Table 2. It is known that the h-index is one of the metric coefficients that could be used for measuring the author's scientific output [44]. It is seen that Scopus and WoS point out different authors as impactful in the investigated area of AI application in teaching. The first most successful authors according to Scopus are: Wang Y., Wang X. and Hwang G-J and considering WoS data are: Chai C. S., Li J., and Ahmad S. F.

**Table 2.** The author's impact (h-index) according to Scopus and Web of Science.

| Scopus | | Web of Science | |
|--------|---------|----------------|---------|
| **Author** | **H-Index** | **Author** | **H-Index** |
| WANG Y | 11 | CHAI CS | 3 |
| WANG X | 8 | LI J | 3 |
| HWANG G-J | 7 | AHMAD SF | 2 |
| LIU Y | 7 | ALAM MM | 2 |
| YANG Y | 7 | CHEN L | 2 |
| ZHANG J | 7 | CHEN LJ | 2 |
| ZHANG X | 7 | CHEN Y | 2 |
| ALAM A | 6 | CHIU TKF | 2 |
| CHEN Y | 6 | CUI XW | 2 |
| LIU J | 6 | DAI DD | 2 |

*3.2. Determining the Impact of Scientific Publications*

Through the use of citations as usage and visibility indicators, we were able to evaluate the composition of the core body of basic literature on AI in teaching with respect to significant papers, writers, and countries. In addition, citation context analysis provided insightful viewpoints on the contributions made by certain academics and research teams, as well as the influence that citations have on researchers' output.

Figure 5 displays a compilation of the most frequently cited countries, with China notably leading, a trend attributed to its substantial volume of publications.

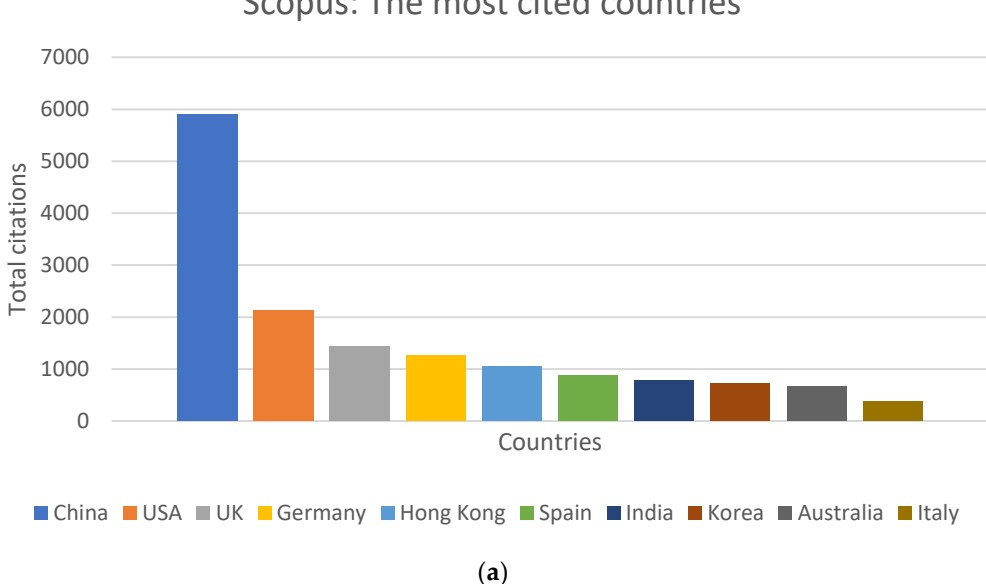

(**a**)

**Figure 5.** *Cont.*

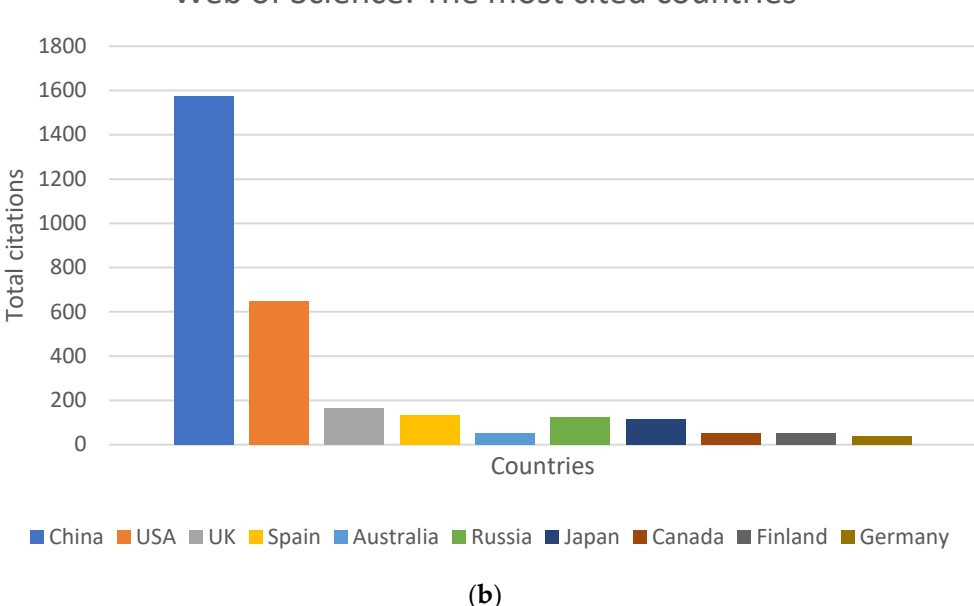

(**b**)

**Figure 5.** The most cited countries (total citations) according to (**a**) Scopus, (**b**) Web of Science.

Other most cited countries (total citations) concerning data from Scopus are: the USA, United Kingdom, Germany, Hong Kong, Spain, India, Korea, Australia and Italy. The average article citations (AAC) is bigger for Vietnam = 56, and followed by articles with authors from Hong Kong = 23.91, Czech Republic = 19.53, Estonia = 17.50 and Netherlands = 16.86. In the middle in this ranking are Finland = 15.18 and Pakistan = 14.00. Smaller AAC has Germany = 13.43, Fiji = 13.00 and UK = 12.63.

In Web of Science, in addition to countries like China, the USA, the UK, Spain, Germany, and Australia, which are among the most cited nations according to Scopus, there are also entries for Finland, Russia, Japan, and Canada. Japan is the country with bigger average article citations of 116, followed by USA = 20.97, Russia = 17.86, UK = 16.70, Finland = 13.25, Estonia = 10.00, Pakistan = 9.50, Belgium = 9.00, Poland = 9.00 and Spain = 8.93.

Browsing the most cited documents provided a very quick understanding of "AI in teaching" mainstream related research and its major trends. The most globally cited documents according to Scopus and Web of Science are presented in Table 3.

**Table 3.** The globally cited documents according to Scopus and Web of Science.

| Paper | DOI | Total Citations | TC per Year | Normalized TC |
|---|---|---|---|---|
| **According to Scopus** | | | | |
| Zawacki-Richter, O.; Marín, V.I.; Bond, M.; Gouverneur, F., 2019, International Journal of Educational Technology in Higher Education [45] | 10.1186/s41239-019-0171-0 | 606 | 121.2 | 58.62 |
| Chen, L.; Chen, P.; Lin, Z., 2020, IEEE Access [46] | 10.1109/ACCESS.2020.2988510 | 326 | 81.5 | 42.71 |
| Dwivedi, Y.K.; Kshetri, N.; Hughes, L.; Slade, E.L.; Jeyaraj, A.; Kar, A.K.; Baabdullah, A.M.; Koohang, A.; Raghavan, V.; Ahuja, M.; et al., 2023, International Journal of Information Management [47] | 10.1016/j.ijinfomgt.2023.102642 | 300 | 300 | 150.8 |
| Smutny, P.; Schreiberova, P., 2020, Computers & Education [48] | 10.1016/j.compedu.2020.103862 | 258 | 64.5 | 33.8 |
| Kasneci, E.; Seßler, K.; Küchemann, S.; Bannert, M.; Dementieva, D.; Fischer, F.; Gasser, U.; Groh, G.; Günnemann, S.; Hüllermeier, E.; et al., 2023, Learning and Individual Differences [49] | 10.1016/j.lindif.2023.102274 | 257 | 257 | 129.19 |

**Table 3.** *Cont.*

| Paper | DOI | Total Citations | TC per Year | Normalized TC |
|---|---|---|---|---|
| Xie, H.; Chu, H.-C.; Hwang, G.-J.; Wang, C.-C., 2019, Computers & Education [50] | 10.1016/j.compedu.2019.103599 | 240 | 48 | 23.22 |
| Hwang, G.-J.; Xie, H.; Wah, B. W.; Gašević, D., 2020, Computers and Education: Artificial Intelligence [51] | 10.1016/j.caeai.2020.100001 | 237 | 59.25 | 31.05 |
| **According to Web of Science** | | | | |
| Hashimoto, D. A.; Rosman, G.; Rus, D.; Meireles, O. R., 2018, Annals of Surgery [52] | 10.1097/SLA.0000000000002693 | 388 | 64.67 | 12.62 |
| Chen, L.; Chen, P.; Lin, Z., 2020, IEEE Access [46] | 10.1109/ACCESS.2020.2988510 | 176 | 44 | 12.37 |
| Chassignol, M.; Khoroshavin, A.; A Klimova, A.; Bilyatdinova, A., 2018, Procedia Computer Science [53] | 10.1016/j.procs.2018.08.233 | 116 | 19.33 | 3.77 |
| Brock, J. K.-U.; von Wangenheim, F. 2019, California Management Review [54] | 10.1177/1536504219865226 | 116 | 23.2 | 12.12 |
| Sit, C.; Srinivasan, R.; Amlani, A. et al., 2020, Insights Imaging [55] | 10.1186/s13244-019-0830-7 | 105 | 26.25 | 7.38 |

The most cited paper considering normalized total citations (NTCs) = 150.8 is an opinion paper: "So what if ChatGPT wrote it"? Multidisciplinary perspectives on opportunities, challenges and implications of generative conversational AI for research, practice and policy", which is open access and is published in *International Journal of Information Management* [47]. The paper expresses the opinion of 43 experts in different areas, pointing out positive and negative issues regarding the usage and impact of generative artificial intelligence technology. The corresponding author is from Swansea University, Swansea, United Kingdom, and other authors are from different countries. The second most cited article (NTC = 129.19) has the topic "ChatGPT for good? On opportunities and challenges of large language models for education", with multiple authors and published in *Learning and Individual Differences* journal [49]. The authors discus some possible applications of large language models in support of educators and learners, as well as address several challenges and opportunities. The university of the corresponding author is Technical University of Munich, Germany, and the rest of the authors possess affiliations of German universities. One other influential paper (NTC = 58.62) is "Systematic review of research on artificial intelligence applications in higher education—where are the educators?", written by Zawacki-Richter et al. and published in *International Journal of Educational Technology in Higher Education* [45]. The authors talk about the need for a better connection between pedagogy and the application of artificial intelligence in higher education, as well as point out some risks and ethical implications. All authors of this paper are from the University of Oldenburg in Germany.

### 3.3. Tracing Collaboration Patterns

The following stage involved conducting a co-authorship analysis to evaluate the patterns in collaboration across authors, countries, and institutions, as well as to gauge the scientific impact of individual scholars. Answers to fundamental queries like "Who collaborates with whom and how?", "Does collaboration truly influence the impact?", "Can past collaborations be assessed?", "What happens with collaborations between industries and universities?", etc., are typically sought after in co-authorship analyses.

Country collaboration networks are presented in Figure 6. According to the data from Scopus, three big clusters are formed with well-developed collaboration among countries. In the first cluster (in red), the authors from China most often collaborate with authors from the USA, India, Hong Kong, Korea, Thailand, and Australia. The stronger collaborative connections in the second cluster (in green) are among the UK, Germany, Sweden, France, Italy, and others. In the third cluster (in blue), authors from Spain, Portugal, Brazil, Colombia, Ecuador, Mexico, Peru, and Chile work collaboratively on the investigated

topic. According to the Web of Science, several smaller clusters are formed. China, India, Thailand, Korea, Japan, Switzerland, and Denmark form the bigger cluster (in orange). UK, Canada, Austria, Israel, Mexico, and Brazil form another cluster (in blue). The authors from the USA, Australia, Germany, Singapore, Spain, and Chile are in strong collaboration, forming a single cluster (in brown).

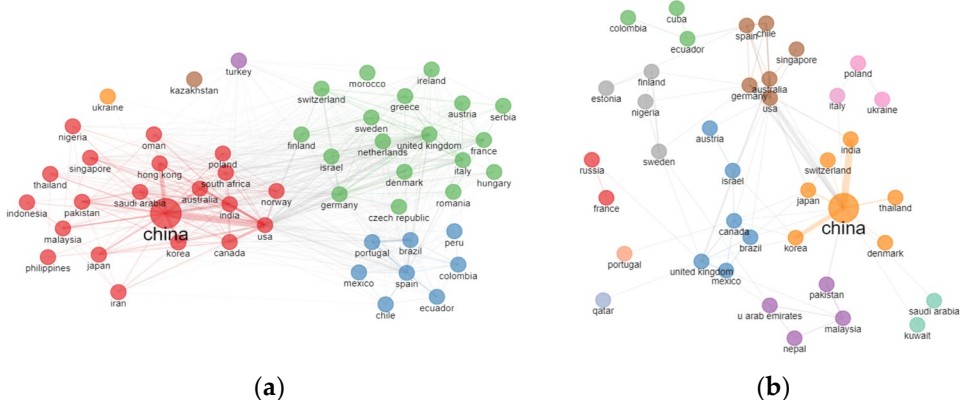

(**a**)  (**b**)

**Figure 6.** Country collaboration networks according to (**a**) Scopus, (**b**) Web of science.

The institution collaborative network map is presented in Figure 7. According to Scopus, the biggest cluster highlights the collaboration among Beijing Normal University, Shanghai Jiao Tong University, University of Cambridge, and The Chinese University of Hong Kong (in red). What is more important here is that the collaboration can be seen both between authors from the same country but different institutions, and between authors from different countries and institutions. This indicates a globalization of the issue and an expanding network of scientists and scholars who are outlining educationally relevant problems regarding the uptake and use of AI by educators and learners.

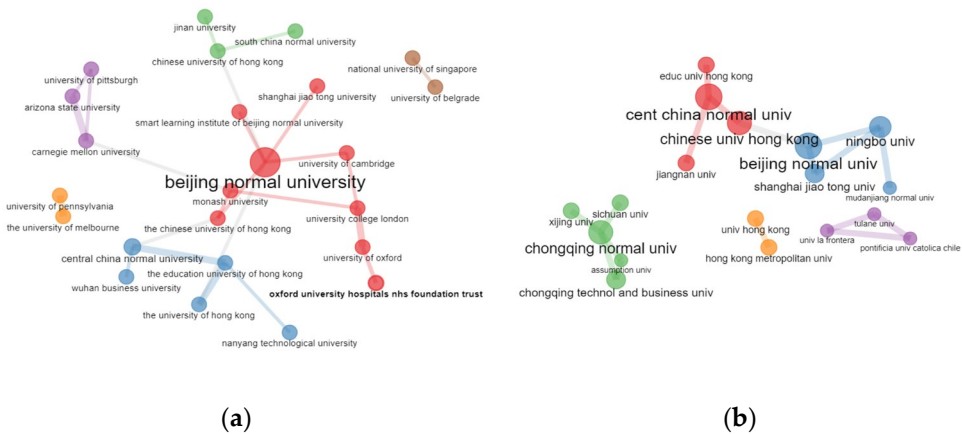

(**a**)  (**b**)

**Figure 7.** Institutions collaborative network concerning data from (**a**) Scopus, (**b**) Web of Science.

The most influential authors are also mapped as shown in Figure 8. It can be observed that the constructed map according to Scopus is very complex, outlining multiple connections not only within entire the cluster, but also outside it. For example, Wang Y. is strongly connected not only with authors from his cluster, but also the author is in relationships with authors from all other clusters. The map created from data of Web of Science is organized through several smaller clusters, as these clusters are not connected among themselves. This means that authors work in isolation by forming separate research groups.

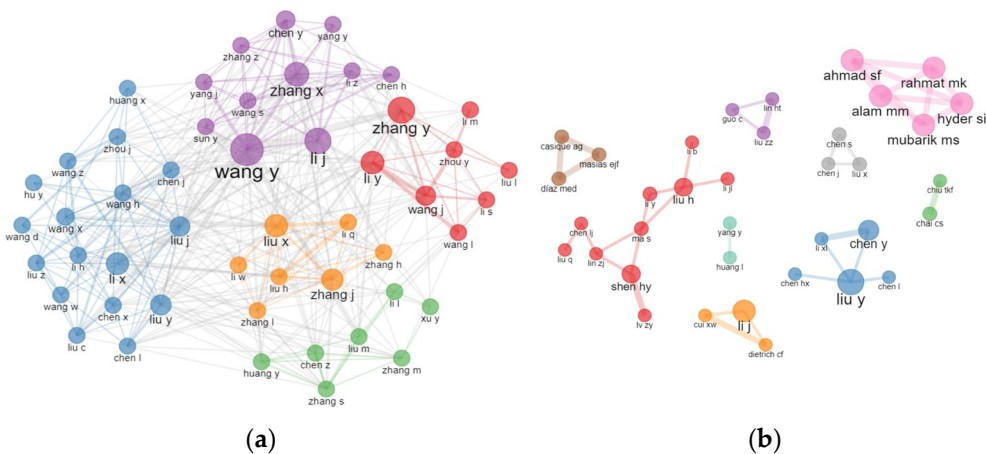

(**a**)          (**b**)

**Figure 8.** Collaboration among the most influential authors: (**a**) Scopus, (**b**) Web of Science.

### 3.4. Identifying Key Research Areas and Emerging Trends

In order to identify the themes that are most commonly discussed among academia and the topics that are representative for research in the period 2018–2023, we looked at the author' keywords, extracted by Biblioshiny. They are similar in Scopus and Web of Science, and could be classified into three groups: (1) *used techniques* like artificial intelligence, machine learning, and deep learning; (2) *popular technologies*: ChatGPT, learning analytics, and virtual reality; and (3) *educational context*: teaching, higher education, active learning, e-learning, and online learning. The most frequent keywords used by authors during 2023 with the purpose of better describing the paper content are shown in Figure 9.

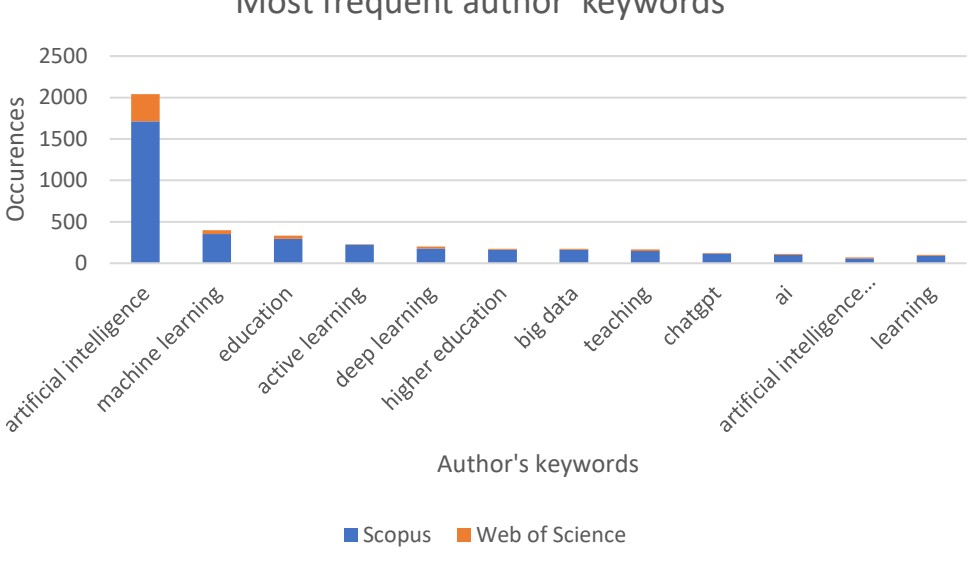

**Figure 9.** Most frequent author keywords concerning bibliometric data during 2023 from Scopus and Web of Science.

The graphics in Figure 10 present trend topics of author keywords during the investigated interval of 2018–2023, with the chart size indicating how many times a given keyword is used by authors. It can be seen that for 2023, the program points out the following as trend topics according to Scopus: *ChatGPT*, *medical education*, *educational technology*, *artificial intelligence*, *machine learning*, and *education*; and according to Web of Science: *ChatGPT*, *e-learning*, *bibliometric analysis*, *teaching*, and *artificial intelligence*. Obviously, ChatGPT is a technology that is gaining attention in teaching and in education as a whole, and will be in focus for discussion and further investigation in the future. The increasing role of artificial intelligence and machine learning in teaching practice is also visible.

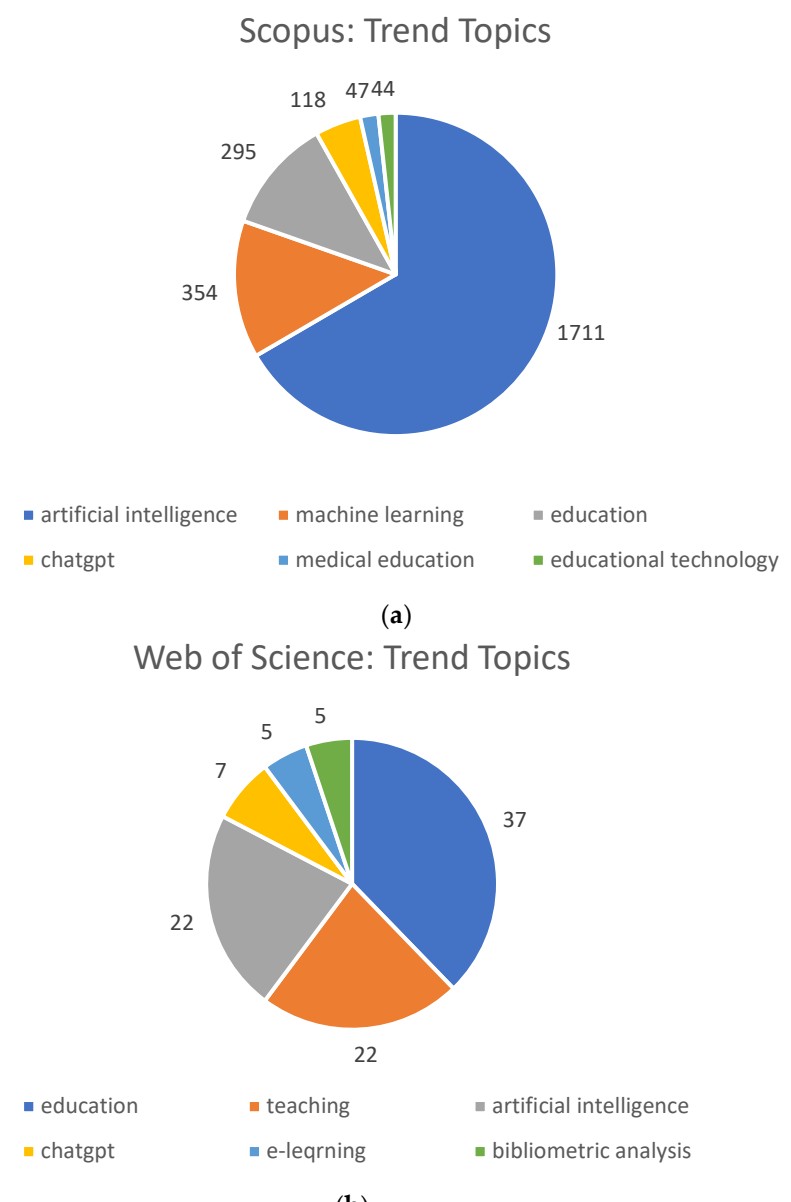

**Figure 10.** Trend topics concerning (**a**) Scopus, (**b**) Web of Science.

The co-occurrence network with 50 nodes (Figure 11) presents grouped words in a cluster concerning a given topic. Two bigger clusters are observed: (1) In the first cluster (in red color) with the highest frequency of occurrence is the term *artificial intelligence*, which is connected to terms like *machine learning*, *deep learning*, *teaching*, *education*, *ChatGPT*, *learning analytics*, *neural networks*, *chatbot*, *personalized learning*, *educational technology*, and *others*. (2) In the second cluster (in purple), the included terms are *big data*, *augmented reality*, *virtual reality*, *internet of things*, *teaching reform*, *data mining*, *smart education*, *cloud computing*, and *others*. (3) The next cluster (in blue) is formed around the keywords *higher education*, *e-learning*, *online learning*, *COVID-19*, *innovation*, and *engineering education*. (4) The terms *active learning*, *blended learning*, and *gamification* (in green) form another cluster. The last cluster is marked with one term, *flipped classroom* (in brown). In Web of Science, the main cluster is similar to the biggest cluster in Scopus and is formed around the term *artificial intelligence* (in green). The different clustering could be explained by the documents that are indexed in the two databases Scopus and Web of Science, which in most cases differ in terms of quantity and published sources, despite the similar nature of the discussed and researched problems.

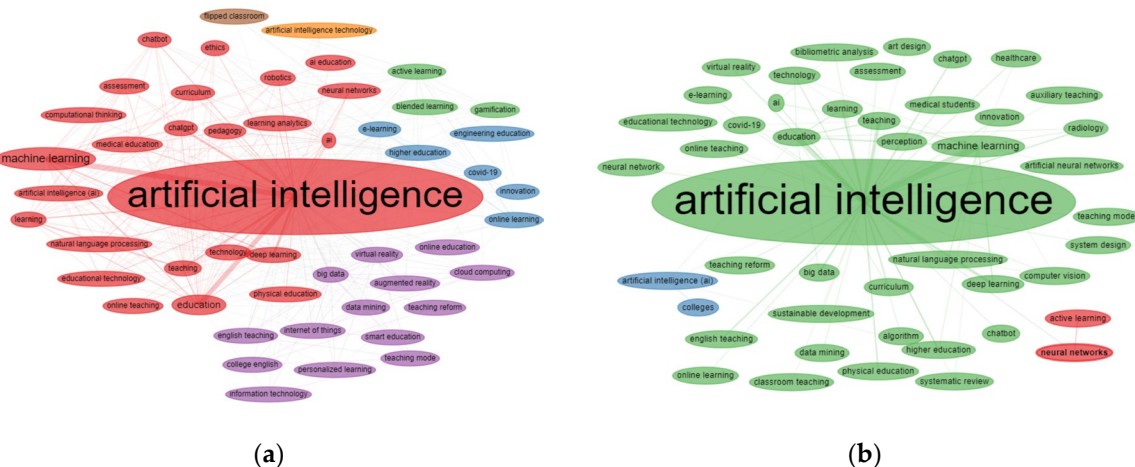

**Figure 11.** Co-occurrence network of author's keywords considering bibliometric data from (**a**) Scopus, (**b**) Web of Science.

## 4. Discussion

This increase in the number of published articles in the last 5 years shows that the topic is extremely relevant, researched, and discussed, as well as of great importance to the field of education, specifically in the context of teaching. Authors from China, USA, Spain, India, UK, and Germany are world leaders, standing out with the largest number of scientific publications, as well as with the highest impact according to Scopus and WoS. Authors from countries such as Australia, Brazil, Italy, and Malaysia according to Scopus, and Canada, Chile, Ecuador, and Korea according to WoS, also fall into the top ten. The scientific output of authors from China stands out significantly compared to the production of authors from other countries. The largest number of articles devoted to the researched topic, according to Scopus, are published in the *Journal of Physics: Conference Series* (SJR 2022: 0.18) and, according to the WoS, *Journal of Intelligent and Fuzzy Systems* (Q2, SJR 2022: 0.37). The articles of the top ten authors are most often published in journals and book series in the fields of computer science, communications, and intelligent systems.

The top three author universities with the highest number of publications according to Scopus are in China: Beijing Normal University (China), Central China Normal University (China), and South China Normal University (China); and according to WoS are in China, Canada, and Hong Kong: Chinese University of Hong Kong (Hong Kong), McGill University (Canada), and Beijing Normal University (China).

The most productive and significant author, shown by h-index = 11, is Wang Y. concerning data from Scopus, and, according to the WoS, the authors Chai C. S. and Li J. with h-index = 3 are the most influential.

The countries of the most cited authors according to both Scopus and WoS indexing databases are China, the USA, and the UK, with China significantly dominating. According to the AAC (average article citation) coefficient, the most cited authors are from Vietnam, Hong Kong, and the Czech Republic (according to Scopus) and Japan, USA, and Russia (according to WoS).

Considering the NTC (normalized total citation) coefficient, three articles stand out with respective titles: "So what if ChatGPT wrote it? Multidisciplinary perspectives on opportunities, challenges and implications of generative conversational AI for research, practice and policy", "ChatGPT for good? On opportunities and challenges of large language models for education", and "Systematic review of research on artificial intelligence applications in higher education–where are the educators?", showing the position of researchers, educators, and experts on the application of ChatGPT, AI and large language models in education and the resulting challenges and implications. The authors of the first paper number 43 and are from different countries, with a corresponding author from the UK, showing a powerful collaboration in research on the application of artificial intelligence

and its influence in various fields. The authors of the second article are from universities in Germany, bringing together their experiences and observations to discuss how ChatGPT and large language models influence education. The third article is written by authors from a university in Germany, specifically interested in the future of higher education in an environment of increasing numbers of AI applications, and the role of the educator in this situation.

The constructed collaborative networks show how a part of the research teams work in isolated groups, and another part demonstrates a wide connectivity, not only between scientists from different institutions of one country, but also between scientists from different countries and institutions. This shows that the authors of the collaborative teams, who are from different institutions and/or countries, see in the same way or unify their notions and knowledge in the created situation regarding the emerging problems for teachers and learners when using AI in education. Authors from different parts of the world see the advantages and disadvantages of AI for education, giving recommendations or possible solutions.

The identified key areas of research depending on the keywords used by the authors can be classified into three groups depending on the techniques used in educational applications, such as *artificial intelligence*, *machine learning*, and *deep learning*; applicable technologies like *ChatGPT*, *learning analytics*, and *virtual reality*, and the context of the application of these techniques and technologies in education: *teaching*, *higher education*, *active learning*, *e-learning*, and *online learning*.

If we combine the Scopus and WoS trending topics, it can be said that *ChatGPT* and *AI* are the topics that will be intensively discussed from different points of view: educational, technological, pedagogical, psychological, ethical, and legal, showing benefits for all educational participants, as well as all risks and negative influence on educational practice.

Summarized information regarding the investigated scientific production of the query "artificial intelligence" and teaching, applied in Scopus and Web of Science, is presented in Table 4.

**Table 4.** Summarized information according to Scopus and Web of Science.

| Scientific Database/Parameter | Scopus | Web of Science |
|---|---|---|
| Timespan | 2018–2023 | 2018–2023 |
| Sources | 2092 | 289 |
| Documents | 6010 | 500 |
| Annual Growth Rate | 25.42% | 39.33% |
| Authors | 12,973 | 1338 |
| Authors of single-author | 1112 | 152 |
| International co-authors | 13.39% | 14.6% |
| Co-authors per doc | 3.01 | 2.87 |
| Author's keywords | 11,327 | 1114 |
| References | 157,943 | 14,115 |
| Document average age | 2.82 | 2.7 |
| Average citation per doc | 4.841 | 6.49 |

Figure 12 presents the obtained "big picture" taking into account the formulated main objectives of this work in the explored four groups: establishing a descriptive structure of the scientific production, determining the impact of scientific publications, tracing the collaboration patterns, and identifying the key research areas and emerging trends.

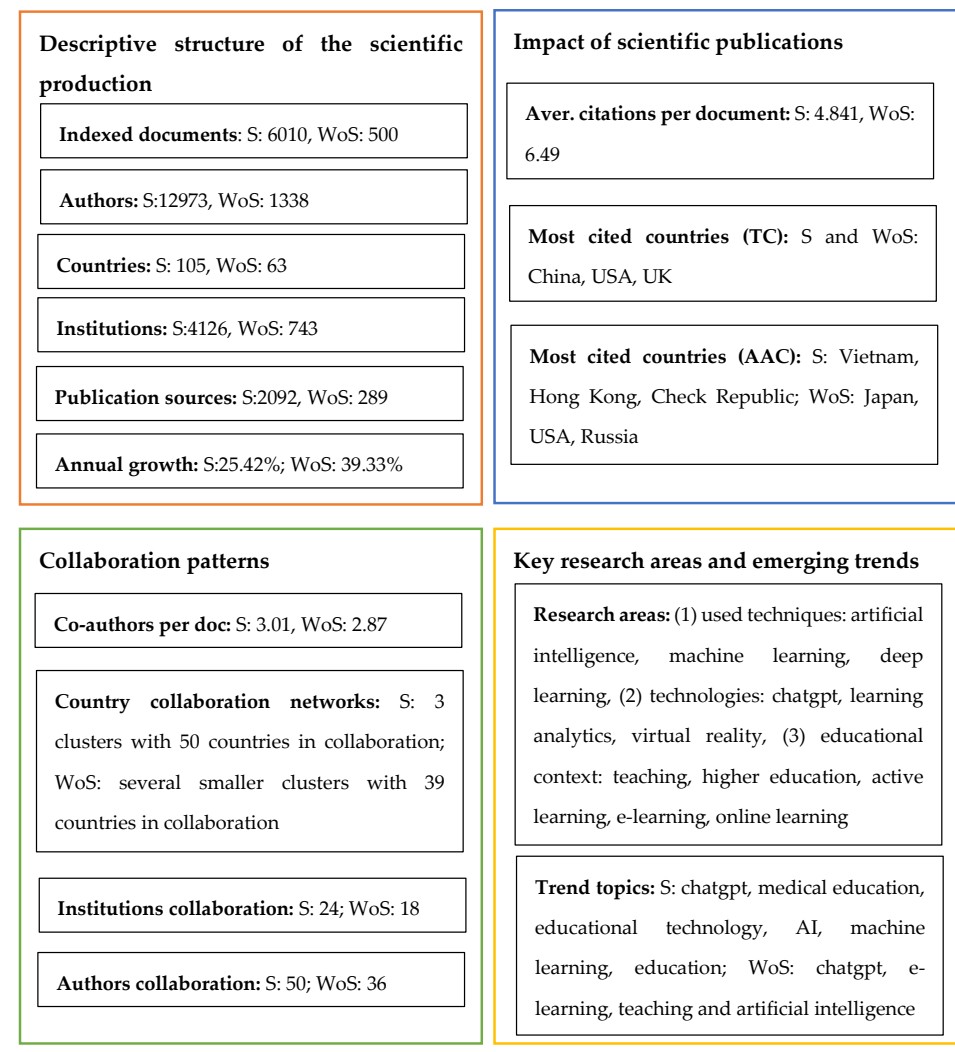

**Figure 12.** Summarized information considering the main objectives.

Although the data we analyzed provided interesting insights on research in AI in teaching, there are some limitations that should be acknowledged. Initially, our investigation relied on a sample of documents sourced from Scopus and WoS. However, there are also additional databases that catalog papers on AI in teaching, and it is advisable to explore these alternatives in the future. Secondly, it is widely recognized that our sample predominantly features documents published in English. While both Scopus and the WoS Core Collection do include non-English journals or literature (with translations of non-English titles into English), the volume of publications in other languages is considerably lower. Thirdly, through an examination of the dataset spanning the last 5 years, we acquired a comprehensive overview of the AI in teaching field, recognizing that the influence of documents, authors, and journals evolves over time. Thus, the chosen timespan (2018–2023) should be reassessed by similar future research. However, these limitations can be seen as challenges that are opening up new avenues for future research.

## 5. Conclusions

This research is based on a bibliometric study of scientific production indexed in Scopus and Web of Science databases, carried out between 2018 and 2023. The main findings increased our understanding of the "AI in teaching" domain: we identified vital research, landmark studies in the development of the field, critical past contributions, emerging trends, and potentially transformative ideas. A summary of the results aligned with the formulated objectives can be encapsulated as follows:

- *Descriptive structure of the scientific production*—It can be said that the topic related to AI in teaching is attracting the attention of more and more research groups, showing its global aspect, important meaning, and need for discussion and further investigation. The findings reveal the increased scientific production during the last 5 years, with annual growth according to Scopus of 25.42%, and 39.33% according to WoS. China's prominent position in both productivity and influence within this domain further accentuates the actual significance of AI in teaching. The increasing scientific production over the last five years, as evidenced by substantial growth rates in Scopus and Web of Science, signals a growing recognition of the subject's relevance and the urgent need for continued research and discourse. The most prolific authors hail from China, USA, Spain, India, UK, and Germany, yet it is evident that research teams from China exert a dominant influence.
- *The impact of scientific publications*—China emerges prominently in this study as a central player, occupying the leading position in both productivity and influence. Furthermore, China stands out among the most cited countries, underscoring its significant impact on the global discourse. The top three most cited articles identify problems related to the usage of ChatGPT and generative conversational AI in different domains, the application of ChatGPT and large language models in education, and the role of educators in the utilization of AI in higher education.
- *Collaboration patterns*—Many research teams are formed as some of them are strongly connected in an international aspect that is confirmed through the constructed collaborative networks. Some research groups work in isolation without any connections outside. A very small share of papers is written by a single author.
- *Key research areas and emerging trends*—It seems that topics related to the terms *ChatGPT, generative AI, large language models, intelligent systems*, and *learning analytics* will define the future research landscape. Such issues like how, how much, and how far to go with AI in teaching are discussed from different points of view: educational, technological, pedagogical, psychological, ethical, and legal, outlining not only the supportive benefits for teachers and learners, but also potential risks and challenging problems.

In essence, our study affirms that AI in teaching is not merely a technological advancement but also a transformative force that requires ongoing attention, collaboration, and ethical consideration. Our research aligns with the conclusions of Markauskaitė et al. [56], which indicate that a comprehensive understanding of capabilities requires a shift away from AI-centric viewpoints and consideration of the technology, cognition, social interaction, and values ecosystem. As we navigate this evolving landscape, understanding and harnessing the potential of AI in education will be crucial for shaping a future where technology seamlessly integrates with pedagogy to enhance learning outcomes and empower educators worldwide.

**Author Contributions:** Conceptualization, G.G., M.I. and C.H.; methodology, M.I. and G.G.; formal analysis, M.I. and G.G.; investigation, G.G., M.I. and C.H.; resources, G.G., M.I. and C.H; writing—original draft preparation, M.I., G.G. and C.H.; writing—review and editing, G.G., M.I. and C.H.; visualization, M.I. All authors have read and agreed to the published version of the manuscript.

**Funding:** This research was supported by the Bulgarian FNI fund through the project "Modeling and Research of Intelligent Educational Systems and Sensor Networks (ISOSeM)", grant number КП-06-H47/4 from 26 November 2020.

**Institutional Review Board Statement:** Not applicable.

**Informed Consent Statement:** Not applicable.

**Data Availability Statement:** The data presented in this study are available on request from the corresponding author.

**Conflicts of Interest:** The authors declare no conflicts of interest.

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
