# Peer review of "Unveiling Insights: A Bibliometric Analysis of Artificial Intelligence in Teaching"

_informatics, doi:10.3390/informatics11010010_

Round 1
Reviewer 1 Report
Comments and Suggestions for Authors
The work focuses on analysing the influence of artificial intelligence on teaching practice from a systematic review study over the last five years. The results indicate an increase in this type of studies and the participative collaboration of different groups. Furthermore, these studies make use of machine learning and deep learning techniques and the latest technologies with ChatGPT, learning analytics and virtual reality. Specifically, these resources are being used in Higher Education in the field of online learning. The research focuses on the use of ChatGPT, chabots, AI, generative AI, machine learning, emotion recognition, large linguistic models, neural networks and decision theory.
It is an interesting, well-structured and rigorous work. However, I would like to make some suggestions for improvement:
Major changes
Revise the introduction and increase the number of citations, especially updated ones, since this is a systematic review study and only 28 are presented in the Bibliographical References section.
2. Before the Methodology section, define the objectives and hypotheses of the study.
3. The conclusions section should be structured on the basis of the answers to the research questions or hypotheses made a priori.
Minor changes
3. Adjust the Figures and tables to the journal's standards.
4. Revise the bibliographical references in the bibliographical references section and bring them into line with the journal's standards.
Author Response
Dear Reviewer, thank you so much for valuable comments and suggestions for paper improvement. We consider all of them in our corrected variant.
Major changes
- Revise the introduction and increase the number of citations, especially updated ones, since this is a systematic review study and only 28 are presented in the Bibliographical References section. - The section Introduction is updated and extended taking into account several other scientific works.
- Before the Methodology section, define the objectives and hypotheses of the study. - The objectives are clearly formulated and precised.
- The conclusions section should be structured on the basis of the answers to the research questions or hypotheses made a priori. - The section Conclusion is improved and structured according to defined objectives.
Minor changes
- Adjust the Figures and tables to the journal's standards. - Some of figures are re-drawn in the form of images and tables to satisfy the journal requirements.
- Revise the bibliographical references in the bibliographical references section and bring them into line with the journal's standards. - The Reference list is checked and corrected.
Reviewer 2 Report
Comments and Suggestions for Authors
- Journal is generally very well written and presented, in addition to the highly relevant context. I found the analysis and results of this journal also very insightful.
- Figure 1 is a good illustration for the PRISMA process
-Rest of the figures are a bit unclear due to the small font of their axes and legends, increase font size or figure size as a whole
-No comments on the analysis conducted
Author Response
Dear Reviewer, thank you so much for your comments.
- Journal is generally very well written and presented, in addition to the highly relevant context. I found the analysis and results of this journal also very insightful. - Thank you so much for this comment.
- Figure 1 is a good illustration for the PRISMA process. - Thank you so much for this comment.
-Rest of the figures are a bit unclear due to the small font of their axes and legends, increase font size or figure size as a whole. - A big part of the figures are re-drawn in the form of images and tables to be more clear and understandable.
-No comments on the analysis conducted. - Thank you so much for this comment.
Reviewer 3 Report
Comments and Suggestions for Authors
Overall Major Comments;
Since this is a bibliometric analysis it may be good to give more information about what you have done in the introduction.
I recommend expanding the bibliometric analysis to provide a more comprehensive understanding of the field. While Sections 3.1 and 3.2 effectively address publication and citation analysis, additional depth could be achieved by:
Add some information in the introduction about
Citation Context Analysis
Influence of Highly Cited Papers
Historical Citation Trends
Comparative Citation Analysis
-Consider adding some more information about scientometric mapping
-About the PRISMA which includes a 27-item checklist you just give a paper about that however it may be good give a reference for the direct source of prisma like http://www.prisma-statement.org/
Minor Improvements
Figures are created more like a infographic or presentation, consider a more appropriate way of showing your results.
Figure 14
The figure is like an inforgraphic with background images not suitable for a research paper, unless the graphic is for a specific purpose use the regular tables.
In Figure 13
Explain the difference between Scopus vs WOS since the clustering is different for similar terms
In Figure 12
The buble size denotes for the quantity of papers but whats the scale, it may be good to give at least a reference , with that view like that its difficult to analytically understand. You can use similar like Figure 11.
Line 164 is orphan
Comments on the Quality of English Language
Grammar and English appear to be generally consistent with academic writing standards. There are no large fragmented sentences or obvious clarity problems that appear significantly. The paper maintains a formal academic tone, using appropriate terminology and phrases for a scientific article.
Author Response
Dear Reviewer, thank you so much for valuable comments and suggestions for paper improvement. We consider all of them in our corrected variant.
Since this is a bibliometric analysis it may be good to give more information about what you have done in the introduction. - The section Introduction is revised and extended.
I recommend expanding the bibliometric analysis to provide a more comprehensive understanding of the field. While Sections 3.1 and 3.2 effectively address publication and citation analysis, additional depth could be achieved by:
Add some information in the introduction about
Citation Context Analysis
Influence of Highly Cited Papers
Historical Citation Trends
Comparative Citation Analysis
-Consider adding some more information about scientometric mapping
The section Introduction is revised and extended.
-About the PRISMA which includes a 27-item checklist you just give a paper about that however it may be good give a reference for the direct source of prisma like http://www.prisma-statement.org/ - The link of PRISMA is included in the paper.
Minor Improvements
Figures are created more like a infographic or presentation, consider a more appropriate way of showing your results. - A big part of figures is re-drawn in the form of images and tables to satisfy the journal requirements.
Figure 14
The figure is like an inforgraphic with background images not suitable for a research paper, unless the graphic is for a specific purpose use the regular tables. - A big part of figures is re-drawn in the form of images and tables to satisfy the journal requirements.
In Figure 13
Explain the difference between Scopus vs WOS since the clustering is different for similar terms. - An explanation is included.
In Figure 12
The buble size denotes for the quantity of papers but whats the scale, it may be good to give at least a reference , with that view like that its difficult to analytically understand. You can use similar like Figure 11. - The figure is re-drawn in another style to be more clear and understandable.
Line 164 is orphan - The lines are checked.